# Comparative Analysis of the Complete Chloroplast Genome Sequences of Three Closely Related East-Asian Wild Roses (*Rosa* sect. *Synstylae*; Rosaceae)

**DOI:** 10.3390/genes10010023

**Published:** 2019-01-03

**Authors:** Ji-Hyeon Jeon, Seung-Chul Kim

**Affiliations:** Department of Biological Sciences, Sungkyunkwan University, Suwon 16419, Korea; theo_sage17@naver.com

**Keywords:** Rosaceae, *Rosa*, section *Synstylae*, *Rosa multiflora*, *Rosa luciae*, *Rosa maximowicziana*, chloroplast genome, genome comparison

## Abstract

Species belonging to *Rosa* section *Synstylae* (Rosaceae) are mainly distributed in East Asia, and represent recently diverged lineages within the genus. Over decades, inferring phylogenetic relationships within section *Synstylae* have been exceptional challenges, due to short branch lengths and low support values. Of approximately 36 species in the section *Synstylae*, *Rosa multiflora*, *Rosa luciae* and *Rosa maximowicziana* are widely distributed in the Sino-Japanese floristic region. In this study, we assembled chloroplast genomes of these three species to compare the genomic features within section *Synstylae*, and to compare with other infrageneric groups. We found that three *Rosa* sect. *Synstylae* species had lost *infA* genes with pseudogenization, and they were almost identical to each other. Two protein-coding gene regions (*ndhF* and *ycf1*) and five non-coding regions (5’*matK*-*trnK*, *psbI*-*trnS*-*trnG*, *rps16*-*trnG*, *rpoB*-*trnC,* and *rps4*-*trnT*) were identified as being highly informative markers. Within three section *Synstylae* chloroplast genomes, 85 simple sequence repeat (SSR) motifs were detected, of which at least 13 motifs were identified to be effective markers. The phylogenetic relationships of *R. multiflora*, *R. luciae* and *R. maximowicziana* could not be resolved, even with chloroplast genome-wide data. This study reveals the chloroplast genomic data of *Rosa* sect. *Synstylae*, and it provides valuable markers for DNA barcoding and phylogenetic analyses for further studies.

## 1. Introduction

The genus *Rosa* L. (Rosaceae) comprises approximately 125–200 species, distributed through the temperate and subtropical regions of the Northern hemisphere [1,2,3]. In the conventional taxonomy, genus *Rosa* is divided into four subgenera (*Hulthemia*, *Rosa*, *Platyrhodon*, and *Hesperhodos*), and the subgenus *Rosa* is further divided into 10 sections (*Pimpinellifoliae*, *Gallicanae*, *Caninae*, *Carolinae*, *Rosa*, *Synstylae*, *Chinenses* [syn. *Indicae*], *Banksianae*, *Laevigatae*, and *Bracteatae*) [1,4,5]. The *Rosa* section *Synstylae* DC. comprises approximately 36 species, which are well-circumscribed with the distinctive synapomorphic characteristic of connate styles into an exserted column [1,5], while the phylogenetic consectionality with section *Chinenses* has been suggested, from several previous studies [2,5,6,7,8]. *Rosa* sect. *Synstylae* species are mostly distributed in East Asia, of which *Rosa multiflora* Thunb. and *Rosa luciae* Franch. & Rochebr. ex Crép (syn. *Rosa wichurana* Crép) are the most well-known species, as ones of the wild ancestors of cultivated roses [9]. *Rosa* sect. *Synstylae* species are known to be recently diverged, based on poor resolutions and low support values from many previous molecular phylogenetic studies [2,6,7,8,10,11]. Even the most recent attempt to reconstruct the phylogeny of section *Synstylae* did not resolve the phylogenetic relationships with several phylogenetic problems, such as species delimitation failure, poor resolutions, and low support values, and inconsistency among gene trees [5]. Another evidence supporting the close species relationships in section *Synstylae* is incomplete reproductive barriers among congeneric species [12,13]. Indeed, the phylogenetic reconstruction of genus *Rosa* has been notoriously difficult, due to low sequence divergence level and hybridization [7]. 

Chloroplasts are the photosynthetic organelles in plant cells, and the chloroplast genomes contain various essential genes for carbon fixation and metabolite synthesis. Most chloroplast genomes range from 120 to 170 kb in size, and have the quadripartite structure, in which two inverted repeats (IRs) are separated by a large single-copy region (LSC) and a small single-copy region (SSC) [14]. Their genomic structure, gene content, gene order, and base composition are known to be highly conserved, especially in IR regions. For decades, various chloroplast genome regions have been identified as useful molecular markers in systematic and population genetic studies [15,16]. Whole-chloroplast genome sequencing, with the advent of high-throughput next-generation sequencing technique, now generates massive sequence data, helping to overcome the previously unresolved relationships. It also provides genomic information (e.g., structure, gene order, and content), mutations in which imply the presence of the critical events in the evolutionary histories.

In the Eastern Sino-Japanese floristic region, Chang et al. [17] suggested three major florae: the Amur flora, the Liaoning–Shandong flora, and the Southern Chinese–Japanese flora. Three representative species of *Rosa* sect. *Synstylae* in this study, i.e., *R. multiflora*, *R. luciae*, and *Rosa maximowicziana* Regel, are widely distributed in the Eastern Sino-Japanese floristic region, and each species represents different floristic elements. *Rosa multiflora* is found in the Liaoning–Shandong flora, while *R. luciae* and *R. maximowicziana* occur in the Southern Chinese flora and the Amur flora, respectively (Figure 1). Though their distributions are to some extent, parapatric (between *R. maximowicziana* and *R. multiflora*/*R. luciae*) or to overlap a large portion (between *R. multiflora* and *R. luciae*), these species are distinct in the ecological, morphological, or phenological characteristics (Table 1). While they have the evident differences in these characteristics, their species relationships or delimitations were not clear with molecular data in previous studies, suggesting short evolutionary history on them [2,5,6,7,10,11].

In previous studies, only several chloroplast region data were used to infer the phylogeny of *Rosa* sect. *Synstylae* [5,6,7,8,10]. In this study, whole chloroplast genome sequences of three *Rosa* sect. *Synstylae* species (*R. multiflora*, *R. luciae*, and *R. maximowicziana*) were sequenced to (1) understand the chloroplast genomic features of *Rosa* sect. *Synstylae*; (2) evaluate interspecific divergence among the chloroplast genomes of *Rosa* sect. *Synstylae* species; and (3) infer the phylogenetic relationship based on the chloroplast genomic data. In addition to the intra-sectional comparison chloroplast genomes of *Rosa* sect. *Synstylae*, the intra-generic comparison including chloroplast genomes of species in the different infrageneric group was conducted to obtain further insights on chloroplast genomes in genus *Rosa*. This study will provide the information of effective chloroplast markers to identify *Rosa* species, and the insight on directions of further systematic studies on *Rosa* sect. *Synstylae*. Moreover, it will be one of the good model study on the chloroplast genome evolution of very recently diverged species, which are geographically, ecologically, morphologically, or phenologically distinct.

## 2. Materials and Methods

### 2.1. Taxon Sampling and Genome Sequencing

Fresh young leaf samples were collected from wild individuals of three *Rosa* sect. *Synstylae* species (*Rosa multiflora*, *R. luciae* and *R. maximowicziana*; Table 2). Voucher specimens for three *Rosa* accessions were deposited in Ha Eun Herbarium, Sungkyunkwan University (SKK). Total genomic DNA was extracted using Exgene Plant SV mini kit (Geneall, Seoul, Korea) and sequenced using Illumina HiSeq 4000 platform (Illumina, San Diego, CA, USA) at Macrogen (Seoul, Korea), generating paired-end sequence reads.

### 2.2. Chloroplast Genome Assembly and Annotation

The paired-end reads of each species were assembled de novo into the draft chloroplast genome sequence contigs using Velvet 1.2.10 [18]. Assembled draft contigs were imported into Geneious R 10.2.3 (Biomatters Ltd., Auckland, New Zealand) to develop the circular chloroplast genome sequence [19]. Several ambiguous regions (e.g., simple sequence repeat regions) and the junctions between IRs and LSC/SSC regions were confirmed with Sanger sequencing by Polymerase Chane Reaction (PCR) amplification. 

The assembled chloroplast genome sequences were annotated using Geneious. Protein coding genes and ribosomal RNA (rRNA) genes were primarily identified using DOGMA (Dual Organellar GenoMe Annotator) [20], and transfer RNA (tRNA) genes were identified using tRNAscan-SE 2.0 [21]. Further annotations were identified by comparing chloroplast genomes with the well-defined *Nicotiana tabacum* chloroplast genome (Z00044) [22] and *Rosa roxburghii* chloroplast genome (NC_032038) [23] from GenBank as references. Finally, all gene annotations were verified and accurately adjusted via BLASTN and BLASTX searches. Three *Rosa* chloroplast gene maps were drawn with OGDRAW (OrganellarGenomeDRAW) [24]. The annotated chloroplast genomes of three *Rosa* species were submitted to GenBank (Table 2).

### 2.3. Chloroplast Genome Comparative Analyses

The complete chloroplast genomes of *R. multiflora*, *R. luciae*, and *R. maximowicziana* from this study and chloroplast genomes of two other *Rosa* species from other infrageneric groups in GenBank, *Rosa roxburghii* (subgenus *Platyrhodon*; NC_032038) [23] and *Rosa odorata* var. *gigantea* (subgenus *Rosa* section *Chinenses*; KF753637) [25], were compared using mVISTA [26,27] with LAGAN alignment program [28]. To compare the chloroplast genomes both within section *Synstylae* and among infrageneric groups, *R. multiflora* chloroplast genome was used as a reference genome in this analysis. Major variations in gene contents or features of *Rosa* chloroplast genomes were manually identified with Geneious [19]. For the accurate genome comparison, gene annotations of NC_032038 and KF753637 were checked again with BLASTN, BLASTX, and tRNAscan-SE [21].

DNA polymorphisms analysis was performed using DnaSP (DNA Sequence Polymorphism) v6 [29] to calculate the nucleotide diversity (Pi), and to detect highly variable sites among *Rosa* chloroplast genomes. Chloroplast genome sequences were aligned using Geneious Alignment, and the alignment was manually adjusted by using Geneious [19]. Nucleotide diversities were calculated within section *Synstylae* and within genus *Rosa*. In DNA polymorphism analysis, the window length was set to 800 bp, and the step size was set to 200 bp.

### 2.4. Analysis of Simple Sequence Repeats

The simple sequence repeats (SSRs) of 1-6 bp repeats in three chloroplast genomes of sect. *Synstylae* were identified using MISA-web (MIcroSAtellite identification tool-web) [30,31]. The minimum repeat number thresholds were set to 10 for mononucleotide repeats, five for dinucleotide repeats, four for trinucleotide repeats, and three for tetranucleotide, pentanucleotide, or hexanucleotide repeats. The monomeric repeat regions with the repeat number of nine were also counted as SSRs if the same region in other species had 10 or greater.

### 2.5. Phylogenetic Analysis

In the phylogenetic reconstruction of *Rosa* species, five species including three from this study (i.e., *R. roxburghii*, *R. odorata* var. *gigantea*, *R. multiflora*, *R. luciae*, and *R. maximowicziana*) were included. The chloroplast genome of *Fragaria vesca* ssp. *vesca* (NC_015206) [32] and *Dasiphora fruticosa* (NC_036423) [33] from GenBank were used as the outgroups, because the genus *Fragaria* and *Dasiphora* are suggested to be ones of the closely related genera to *Rosa* [34]. The whole chloroplast genome sequences were aligned using Geneious Alignment and the alignment was manually adjusted in Geneious [19].

Maximum likelihood (ML) and Bayesian inference (BI) analyses were conducted to reconstruct the phylogenetic relationships. Tree reconstruction and the bootstrap for ML analysis were performed in IQ-TREE v1.6.1 [35]. The substitution model was selected using ModelFinder [36] in IQ-TREE. ModelFinder selected the three substitution-type models with unequal base frequency and the gamma rate heterogeneity (K3Pu+F+G4) as the best-fitting substitution model, based on Bayesian information criterion (BIC). The phylogenetic tree was reconstructed with IQ-TREE and the branch supports for the tree were estimated with 3,000 bootstrap replicates using UFBoot (Ultrafast Boostrap Approximation) [35,37]. Based on the model test result from ModelFinder, the general time-reversible model with the gamma rate heterogeneity (GTR+G4) was selected as the best-fitting model for Bayesian inference. MrBayes 3.2.6 [38] conducted the Bayesian inference of the *Rosa* chloroplast genomic phylogeny. In Markov chain Monte Carlo (MCMC), a couple of four chains were run for 2,000,000 generations (2 × 2,000,000 generations). The trees were sampled in every 100 generations, and the first 25% of sampling were discarded as burn-in trees. Remaining trees were used to construct a 50% majority-rule consensus tree. 

## 3. Results

### 3.1. General Chloroplast Genome Features

The total genome sequence lengths of three *Rosa* sect. *Synstylae* species (*R. multiflora*, *R. luciae*, and *R. maximowicziana*) were from 156,405 bp (*R. maximowicziana*) to 156,519 bp (*R. multiflora*), and the depth of coverage for chloroplast genome contigs were from 214× (*R. multiflora*) to 1418× (*R. maximowicziana*). Each chloroplast genomes of *Rosa* sect. *Synstylae* species had a typical quadripartite structure, in which a large single-copy region (LSC; 85,529–85,643 bp) and a small single-copy region (SSC; 18,759–18,760 bp) were separated by two inverted repeats (IRs; 26,058 bp) (Figure 2). The gene numbers, gene orders, and GC contents were identical in chloroplast genomes of three *Rosa* sect. *Synstylae* species (Table 3). One hundred and fourteen unique genes were identified in each chloroplast genome of three *Rosa* sect. *Synstylae* species, including 80 protein-coding genes, 30 tRNA genes and four rRNA genes (Table 3). Among the unique genes, 15 genes contained one intron, and two genes contained two introns (Appendix A).

Comparing to the previously identified chloroplast genomes of two *Rosa* species in other infrageneric groups, *R. roxburghii* (subgenus *Platyrhodon*) and *R. odorata* var. *gigantea* (subgenus *Rosa* section *Chinenses*), three *Rosa* sect. *Synstylae* chloroplast genomes (hereafter, the *Synstylae* chloroplast genomes) from this study had smaller genome sizes and fewer genes, whereas the GC contents were identical among five species (Table 3).

### 3.2. Interspecific Comparative Analyses

The interspecific comparison of the gene contents of five *Rosa* chloroplast genomes indicated that *infA* genes of the *Synstylae* chloroplast genomes was pseudogenized by a frameshift mutation (Figure 3). According to the original gene annotations, there are two more differences on gene contents among the *Synstylae*, *R. roxburghii* and *R. odorata* var. *gigantea* chloroplast genomes. First, *trnG(GCC)* gene was missing in *R. roxburghii* chloroplast genome. The other difference was that chloroplast genomes of *R. roxburghii* or *R. odorata* var. *gigantea* contained additional *trnP(GGG)* gene in the region where *trnP(UGG)* gene already exists. We conducted additional BLASTN and tRNAscan-SE searches on these genomes to verify whether these differences are evident. In BLASTN and tRNAscan-SE searches; however, *trnG(GCC)* gene was also found in *R. roxburghii* chloroplast genome. In the other difference, tRNAscan-SE only identified *trnP(UGG)* gene, not *trnP(GGG)* gene. Given these findings, the only difference in gene contents among the *Rosa* chloroplast genomes was that the *Synstylae* chloroplast genomes lost their *infA* genes by pseudogenization, while *R. roxburghii* and *R. odorata* var. *gigantea* chloroplast genomes contained *infA* genes.

Another remarkable difference was the mutations on the 3’ region of *ndhF* genes. In *Rosa* species, two kinds of mutations were found in the 3’ regions of *ndhF* genes. The mutation of T-insertion on the 3’ region of *ndhF* gene of *R. maximowicziana* caused the protein to terminate one amino acid early (746 amino acid residues), compared to the other *Rosa* species (747 amino acid residues). The other mutation was a point mutation on *ndhF* gene of *R. roxburghii*, altering phenylalanine 746 (Phe746) to leucine 746 (Leu746) (Figure 4).

In the overall interspecific chloroplast genome comparison, there was no large differences between *R. multiflora* chloroplast genome and each of the other *Rosa* chloroplast genome (Figure 5). Especially, within section *Synstylae*, the chloroplast genomes of *R. luciae* and *R. maximowicziana* were almost identical to the one of *R. multiflora*, showing very few differences. One distinct difference between the chloroplast genomes of *R. maximowicziana* and *R. multiflora*, however, was that the *R. maximowicziana* chloroplast genome has a 107 bp indel in the *rps4*-*trnT(UGU)* intergenic region. *R. roxburghii* and *R. odorata* var. *gigantea* had more variable sites compared to *R. luciae* and *R. maximowicziana*.

To estimate the genetic distance among *Rosa* chloroplast genomes, nucleotide diversity (Pi) was calculated using DnaSP. The average Pi value for five *Rosa* chloroplast genomes was 0.00154, and the average Pi value for the *Synstylae* chloroplast genomes was 0.00003. IR regions showed lower nucleotide diversity than single copy regions. While a comparison of three *Synstylae* chloroplast genomes was not informative, due to very few variations, a comparison of five *Rosa* chloroplast genome identified several variation hotspots (Figure 6). Six regions showed higher Pi values than other regions (Pi > 0.006). The hotspot with the highest Pi value was the *psbI*-*trnS*-*trnG* region (Pi = 0.01313), followed by four intergenic regions (5’*matK*-*trnK*, *rps16*-*trnG*, *rpoB*-*trnC* and *rps4*-*trnT*), and one gene region (*ycf1*).

### 3.3. Simple Sequence Repeats

MISA identified 85 SSRs in the *Synstylae* chloroplast genomes, including two compound repeat regions (Figure 7 and Appendix A). Except for two compound repeats, 51 mononucleotide, 11 dinucleotide, seven trinucleotide, 11 tetranucleotide, one pentanucleotide, and two hexanucleotide repeats were identified. Both compound repeats were compounds of two monomeric repeat regions. Most SSR regions were identified in single-copy regions, including 69 SSRs in the LSC region and eight SSRs in the SSC region, whereas five SSRs were identified in each IR region (Figure 7). Seventy-five SSRs had the repeat motifs of A/T or A/T combinations. Sixty-five SSRs were located in intergenic regions, 11 SSRs in protein-coding regions, nine SSRs in gene intron regions, and two SSRs in rDNA (Appendix A).

### 3.4. Phylogenetic Analyses

Maximum Likelihood and BI analyses showed identical tree topologies with high support values (Figure 8). *Rosa* sect. *Synstylae* species were monophyletic, strongly supported with a bootstrap value (BS) of 100% in ML, and a posterior probability (PP) of 1.00 in BI. In the chloroplast genome phylogeny, *Rosa odorata* var. *gigantea* was inferred to have a closer relationship with three *Rosa* sect. *Synstylae* species than *R. roxburghii* (100% BS and 1.00 PP support). Within the section *Synstylae*, however, no phylogenetic relationship was inferred, due to short branch lengths and poor support values.

## 4. Discussion

Three chloroplast genomes of *Rosa* sect. *Synstylae* (*Rosa multiflora*, *R. luciae* and *R. maximowicziana*) were de novo assembled for the first time in this study. The *Synstylae* chloroplast genomes were marginally different at best. Not surprisingly, more differences were found between the *Synstylae* chloroplast genomes, and two other chloroplast genomes of *R. roxburghii* (subg. *Platyrhodon*) and *R. odorata* var. *gigantea* (subg. *Rosa* sect. *Chinenses*). The major difference was that the *Synstylae* chloroplast genomes had lost their *infA* gene by pseudogenization (Figure 3). Independent *infA* gene loss is known to be quite common in angiosperm groups [39], but it is remarkable that there was a difference in *infA* gene loss within a genus. Within *Synstylae* species, chloroplast genomes were identical to each other in genome structures, gene contents, and GC contents. They also showed extremely low diversity (Pi = 0.00003), which has been one of major hurdles for the phylogenetic reconstruction in genus *Rosa* [7]. 

Because the *Synstylae* chloroplast genomes had only few variations and no genomic structural differences, several conservative protein-coding regions could be significant candidates for the chloroplast barcoding regions. The *ndhF* gene could be proposed to be one of those, as a frameshift mutation on the 3’ end of *ndhF* gene was identified in *R. maximowicziana*, which caused early termination of translation. This region may be able to be used in the higher taxon, genus *Rosa*, as a missense mutation occurred in the 3’ *ndhF* of *R. roxburghii*. Another candidate is *ycf1* gene (tentatively suggested to be *tic214* [40]), because a conversion of the 543th amino acid (phenylalanine to tyrosine) was identified in *R. maximowicziana*. In the generic level, this region was also suggested as a variation hotspot (Figure 6), showing numerous conservative and missense mutations within genus *Rosa*. These two regions (i.e., *ndhF* and *ycf1* genes) are expected to be highly useful genetic markers in genus *Rosa*, confirming that these regions have been suggested as good markers for DNA barcoding and phylogenetic analyses in Rosaceae [34,41,42,43]. Especially, the *ndhF* gene is known to have a higher substitution rate, and its 3’ region was suggested to be phylogenetically useful, because it shows a high indel frequency, a high AT content and a low homoplasy rate [44]. Indeed, in *Rubus* subg. *Idaeobatus*, which is in the same subfamily with genus *Rosa* (subfamily Rosoideae), the 3’ regions of *ndhF* have shown a wide range of indel variations, resulting in frameshift mutations and alterations on the *ndhF* transcription termination [41]. Moreover, they may provide evolutionary insights on genus *Rosa*, given that they are the protein-coding genes. Because the *Synstylae* chloroplast genomes are highly conserved among species, chloroplast SSRs may provide valuable information for genotyping in the intra- and/or inter-specific level. For SSR genotyping in *Synstylae* species, 13 mononucleotide SSR regions were polymorphic among the species in this study (except for a deletion in the *rps4*-*trnT* region of *R. maximowicziana*). Among those, two regions (*psbH*-*rpoA* and *rpl14*-*rps3* intergenic regions) could differentiate each accession of the *Synstylae* chloroplast genomes (Appendix A). 

*Rosa* sect. *Synstylae* species have been inferred to be very recently diverged, suggesting that *R. multiflora* and *R. luciae* have diverged from their common ancestor ca. 1.6 MYA, based on the chloroplast data [7]. Though *R. maximowicziana* showed two distinctive variations in its chloroplast genome (i.e., the frameshift mutation on the 3’ end of *ndhF* gene and a 107-bp indel in *rps4*-*trnT(UGU)* intergenic region), the *Synstylae* chloroplast genomes were almost identical, showing 0.00003 Pi value among three different genomes (Figure 4, Figure 5 and Figure 6). The comparative and phylogenetic analyses in this study suggested much closer relationships among three *Synstylae* species with lower diversities and shorter branch lengths than previous studies suggested [2,5,7]. The tree topology of sect. *Synstylae* revealed a polytomy, due to the extremely short branch lengths among three species, which suggest marginal phylogenetic differences among them (Figure 8). Given that these species are distinct in the distributions and the ecological, morphological, and phenological characteristics, the incongruence between those characters and the molecular phylogeny may imply the unique evolutionary histories of three *Synstylae* species. The close phylogenetic relationships of three *Synstylae* species may suggest that the evolutionary time was not sufficient for their chloroplast genomes to be distinctly diverged, and/or that they had interspecific genetic introgressions among the *Synstylae* species. Indeed, the incomplete lineage sorting and genetic introgression have been suggested in phylogenetic analyses with chloroplast data of *Synstylae* species [5]. If genetic introgression is occurring in the *Synstylae* clade, chloroplast capture events may occur, which would obscure inferring the species phylogeny. For more phylogenetic resolutions and systematic implications, additional phylogenetic analyses with nuclear markers would be required, because a nuclear genome phylogeny can be different from a chloroplast genome phylogeny [45]. Because the previous studies with various chloroplast markers suggested nonmonophyly of *Rosa multiflora* and *R. luciae* [5,7,8,10], these species might have more than one distinct haplotype. In this study, however, all *Synstylae* species samples were collected in the Korean Peninsula, while they are distributed in the wide range of the Eastern Sino-Japanese floristic region. This sampling bias could affect the chloroplast genome comparative analyses. To verify the existence of various distinct chloroplast haplotypes of *R. multiflora* and *R. luciae*, further phylogeographic and/or population genetic studies, including many accessions from representative sites in the Eastern Sino-Japanese floristic region will shed new light on the evolution of the *Synstylae* species. If there are various distinct chloroplast haplotypes in the section *Synstylae*, and if the haplotype differentiations are found along the geographic gradients, but not along the species boundaries, it is more plausible that the genetic introgressions have occurred, rather than that the chloroplast genome divergence was not sufficient. Several studies found the haplotype sharing in some taxa in which species are closely related, but defined by ecological or morphological differences [46,47,48]. Given that the marginal genome-wide differences in the *Synstylae* species in this study, these species also might share the chloroplast genomes beyond the species boundaries. Therefore, *Rosa* sect. *Synstylae*, including *R. multiflora*, *R. luciae*, and *R. maximowicziana*, can be another model taxonomic group for studying the chloroplast genome evolution of closely related species, which are defined by ecological or morphological differences, but shared the plastomes by the different species.

## Figures and Tables

**Figure 1 genes-10-00023-f001:**
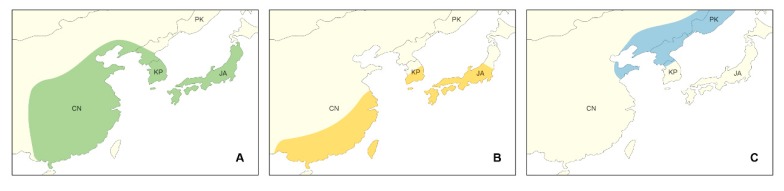
The schematic distribution of three *Rosa* sect. *Synstylae* species. (**A**) *Rosa multiflora* in the Liaoning–Shandong flora; (**B**) *Rosa luciae* in the Southern Chinese–Japanese flora; (**C**) *Rosa maximowicziana* in the Amur flora. CN: China; KP: Korean Peninsula; PK: Primorsky Krai, Russia; JA: Japanese Archipelago.

**Figure 2 genes-10-00023-f002:**
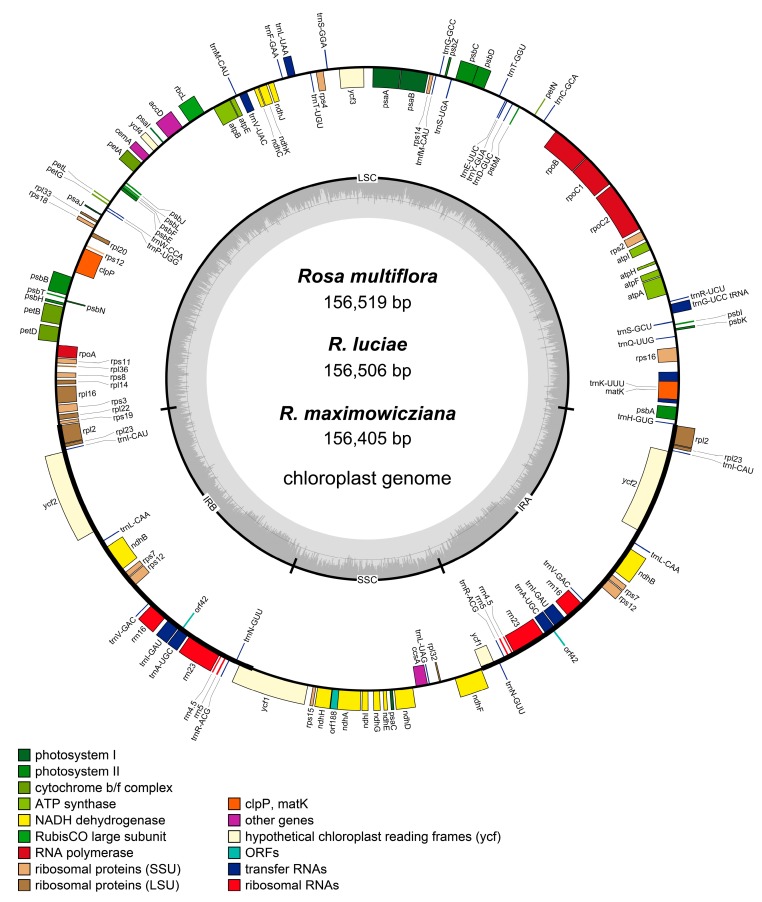
Gene maps of three *Rosa* sect. *Synstylae* species: *R. multiflora*, *R. luciae* and *R. maximowicziana*. The outer circle shows the genes at each locus, and inverted repeat regions are indicated with thicker lines. Genes on the outside of the outer circle are transcribed in a counterclockwise direction, while genes on the inside of the outer circle are transcribed in a clockwise direction. The inner circle indicates the range of the large single-copy region (LSC), the small single-copy region (SSC), and two inverted repeats (IRs), and also shows a GC content graph of the genome. In the GC content graph, the dark gray lines indicate GC content, while light gray lines indicate the AT content at each locus.

**Figure 3 genes-10-00023-f003:**
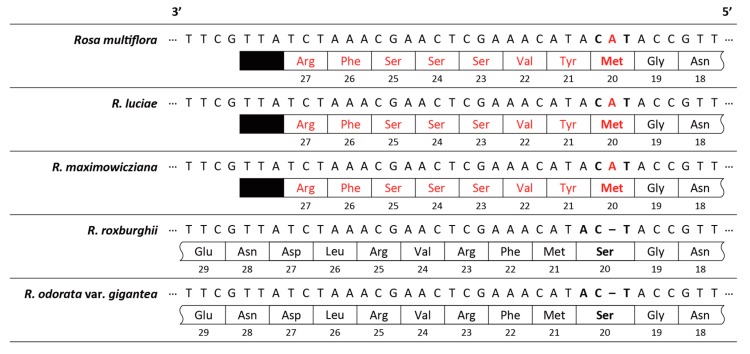
Pseudogenization of *infA* genes in the *Synstylae* chloroplast genomes. In each lane, aligned DNA sequence data are shown on the top. Hyphens indicate indels in the sequences. Boxed ribbons show the amino acid sequences coded from the DNA sequences, and black closed boxes indicate the terminal codons. Numbers under the boxes indicates the amino acid residue numbers. The mutated residues are bolded, and mutations are in red.

**Figure 4 genes-10-00023-f004:**
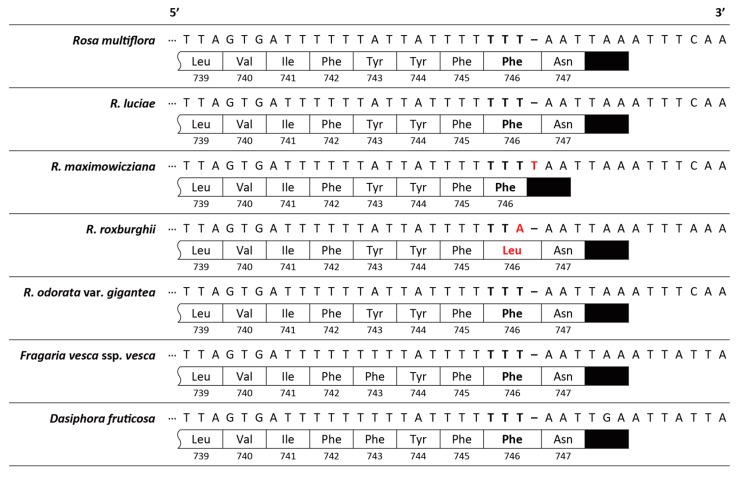
The 3’ region of *ndhF* genes of five *Rosa* species and two other closely related species (*Fragaria vesca* ssp. *vesca* (GenBank accession: NC_015206; [32]) and *Dasiphora fruticosa* (GenBank accession: NC_036423; [33]). In each lane, aligned DNA sequence data are shown on the top. Hyphens indicate indels in the sequences. Boxed ribbons show the amino acid sequences that are coded from the DNA sequences, and black closed boxes indicate the terminal codons. Numbers under the boxes indicate the amino acid residue numbers. The mutated residues are bolded, and mutations are in red.

**Figure 5 genes-10-00023-f005:**
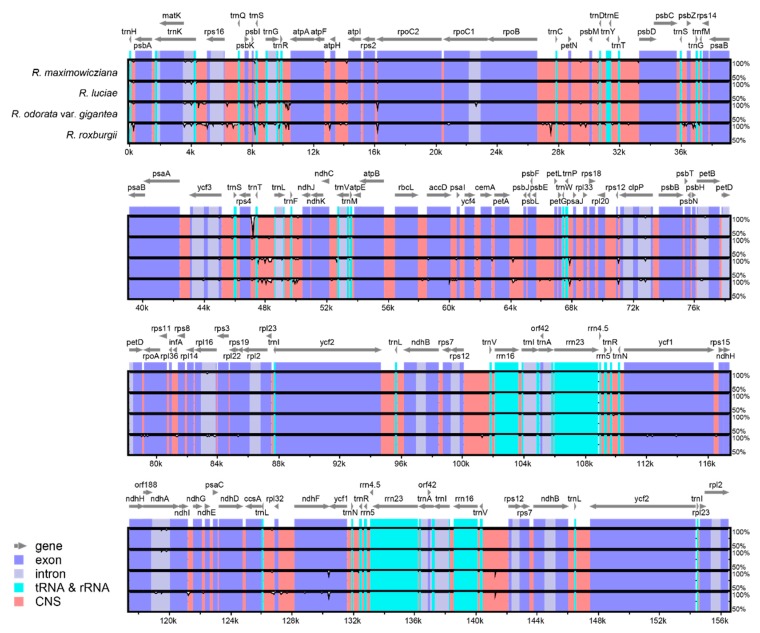
Visualized alignment of the *Rosa* chloroplast genome sequences with annotations, using mVISTA. Each horizontal lane shows the graph for the sequence pairwise identity with a *R. multiflora* chloroplast genome sequence. The x-axis represents the base sequence of the alignment and the y-axis represents the pairwise percent identity within 50–100%. Grey arrows represent the genes and their orientations. Blue boxes represent exon regions and light gray boxes represent intron regions; light blue boxes represent RNA-coding gene regions; red boxes represent non-coding sequence (CNS) regions.

**Figure 6 genes-10-00023-f006:**
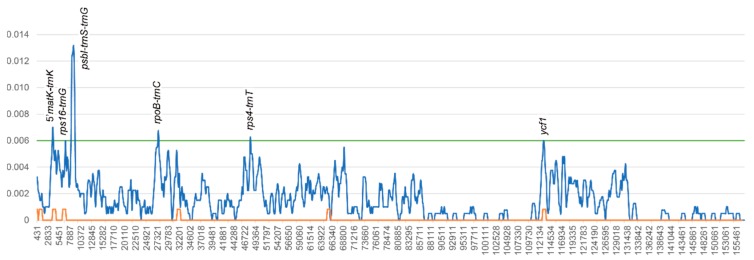
Nucleotide diversity graphs of *Rosa* chloroplast genomes. The x-axis represents the base sequence of the alignment, and the y-axis represents Pi value. The blue line indicates the nucleotide diversity of five *Rosa* chloroplast genomes; the orange line indicates the nucleotide diversity of the *Synstylae* chloroplast genomes; and the green line indicates the threshold for variation hotspots (Pi threshold = 0.006). Each variation hotspot for *Rosa* chloroplast genomes is annotated on the graph.

**Figure 7 genes-10-00023-f007:**
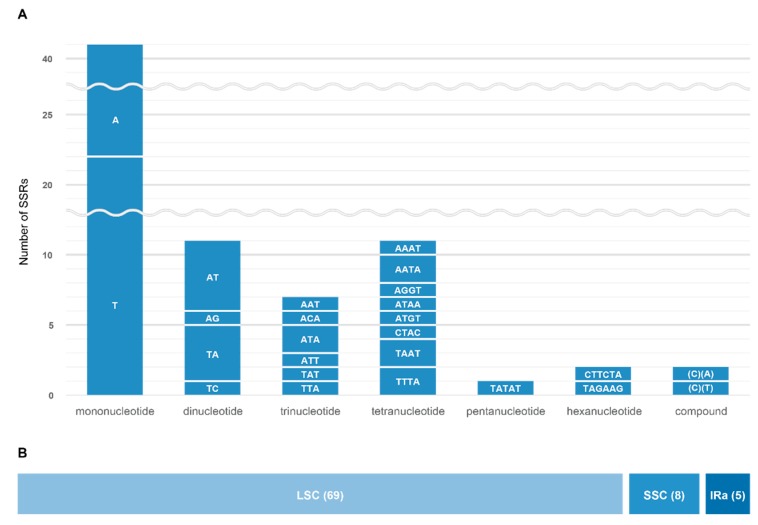
Information of simple sequence repeats (SSRs) of the *Synstylae* chloroplast genomes. (**A**) SSR types and the number of different SSR types. Each repeat motif is indicated in the middle of each bar. (**B**) The proportion of SSR location in the *Synstylae* chloroplast genomes. The locations and SSR numbers were indicated in the middle of bars.

**Figure 8 genes-10-00023-f008:**
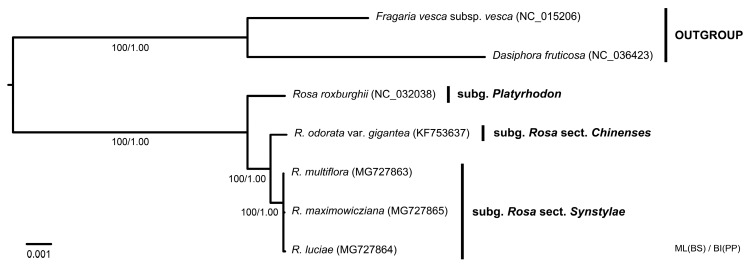
Phylogenetic tree of five *Rosa* chloroplast genomes inferred from maximum likelihood (ML) and Bayesian Inference (BI) based on whole chloroplast genome data. Bootstrap values (BS) from ML analysis and posterior probabilities (PP) from BI were annotated at each branch.

**Table 1 genes-10-00023-t001:** The ecological, morphological, and phenological characteristics of three species of *Rosa* sect. *Synstylae*.

Characteristics	*Rosa multiflora*	*Rosa luciae*	*Rosa maximowicziana*
Habitat	Forest edge	Sea shore/Open space	Thicket/Near the water
Growth habit	Erect	Prostrate	Decumbent
Leaves	Not leathery	Leathery	Not leathery
Stipules	Pectinate	Serrate	Serrate
Prickles	Sparse	Sparse	Dense
Styles	Glabrous	Pubescent	Glabrous
Flowering	April–May	June–July	May–June

**Table 2 genes-10-00023-t002:** Accession information for three *Rosa* species.

Species	Locality	Voucher Number	GenBank Accession
*R. multiflora*	Namyangju, Gyeonggi, Korea	JJH258	MG727863
*R. luciae*	Seoguipo, Jeju, Korea	SGRW001	MG727864
*R. maximowicziana*	Hwaseong, Gyeonggi, Korea	CBRX001	MG727865

**Table 3 genes-10-00023-t003:** The general characteristics of five *Rosa* chloroplast genomes.

Characteristics	*R. multiflora*	*R. luciae*	*R. maximowicziana*	*R. roxburghii*	*R. odorata* var. *gigantea*
GenBank accession number	MG727863	MG727864	MG727865	NC_032038	KF753637
Reference	This study	This study	This study	Wang et al., 2018 [23]	Yang et al., 2014 [25]
Total cpDNA size (bp)	156,519	156,506	156,405	156,749	156,634
LSC size (bp)	85,643	85,631	85,529	85,852	85,767
SSC size (bp)	18,760	18,759	18,760	18,791	18,761
IR size (bp)	26,058	26,058	26,058	26,053	26,053
Number of different genes	114	114	114	115	116 (115 *)
Number of different protein-coding genes	80	80	80	81	81
Number of different tRNA genes	30	30	30	30	31 (30 *)
Number of different rRNA genes	4	4	4	4	4
GC content (%)	37.2	37.2	37.2	37.2	37.2
GC content of LSC (%)	35.2	35.2	35.2	35.2	35.2
GC content of SSC (%)	31.3	31.3	31.3	31.3	31.3
GC content of IR (%)	42.7	42.7	42.7	42.7	42.7

cpDNA: chloroplast DNA; LSC: large single-copy region; SSC: small single-copy region; IR: inverted repeat; * The number of the genes after additional verification.

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
