# Peer review of "Comparative Analysis of the Complete Chloroplast Genome Sequences of Three Closely Related East-Asian Wild Roses (Rosa sect. Synstylae; Rosaceae)"

_genes, 2019, doi:10.3390/genes10010023_

Round 1

Reviewer 1 Report

Using a next-generation sequencing approach, the authors reconstruct the chloroplast genomes of three Rosaceae species, conduct comparative genomic analyses and construct a preliminary phylogeny. They exhaustively describe the genomic differences within and between species groups and mainly conclude that they provide an interesting resource for further comparative studies.

Overall this manuscript falls into the wave of people sequencing chloroplast genomes and publishing them. Consequently this study has to somewhat stick out and make a difference to the many similar studies. I think this is possible with the data presented, however, it requires an improved introduction and discussion. That being said, the study is technically thorough. Lastly, the English language could be improved throughout the manuscript as, for example, articles are often missing.

The introduction lacks a general background about the study species – given that there are very few differences between nominal species (Table 2), my question is – are these actually species? Where do they occur? Do their ranges overlap? Are they ecologically different.

Also, provide some more information about how this study could be of interest beyond people working on Rosaceae.

Lines 54-56: Do not finish the sentence with etc, but explain why these features are of interest.

Lie 118: Indicate why you picked exactly those two species as outgroups. Both are very far away and thus it is not surprising that the phylogeny resolves these well.

Line 152: Genome sized were nearly identical, differing by less than 400bps (Table 2)

Line 172: There is no “confirmation” but rather “findings”. Given that only one individual was sequenced, it remains unknown how variable the chloroplast genomes are within a species.

Line 263: They are marginally different at best

Generally the discussion could include a statement about the broader impact of the results. Given the marginal difference among species as shown in Table 2 and in the phylogeny, the authors should discuss to which degree their species are actually species.

Author Response

Reviewer #1

Comments and Suggestions for Authors

Using a next-generation sequencing approach, the authors reconstruct the chloroplast genomes of three Rosaceae species, conduct comparative genomic analyses and construct a preliminary phylogeny. They exhaustively describe the genomic differences within and between species groups and mainly conclude that they provide an interesting resource for further comparative studies.

Overall this manuscript falls into the wave of people sequencing chloroplast genomes and publishing them. Consequently this study has to somewhat stick out and make a difference to the many similar studies. I think this is possible with the data presented, however, it requires an improved introduction and discussion. That being said, the study is technically thorough. Lastly, the English language could be improved throughout the manuscript as, for example, articles are often missing.

We complemented the introduction and the discussion parts based on your comments and suggestions. Furthermore, the English language was edited by the professional language editing company (Editage, a division of Cactus Communications: http://www.editage.co.kr). The proof of language editing can be provided upon request.

The introduction lacks a general background about the study species – given that there are very few differences between nominal species (Table 2), my question is – are these actually species? Where do they occur? Do their ranges overlap? Are they ecologically different.

We added the general backgrounds of three Rosa sect. Synstylae species in the last paragraph of the introduction. We indicated that they are occurring in the Eastern Sino-Japanese floristic region, and that each species occurs in the different florae of this region. Their distributions are specified on Figure 1, to help understand. To represent that they are distinct species, their ecological, morphological and phenological differences were described in Table 1.

Also, provide some more information about how this study could be of interest beyond people working on Rosaceae.

We focused the attraction point of our study on the genome-wide similarity of three distinct species in spite of their ecological and morphological differences. We presented this point on the last sentence of the introduction. In addition to the introduction, we discussed on this point at the last part of the discussion. By including several case studies that shows the similar pattern to this study, we suggested that Rosa sect. Synstylae species can be another case of the chloroplast sharing. This point could be of interest to the researchers who study on the closely related species, not restricted on researchers working on Rosaceae.

Lines 54-56: Do not finish the sentence with etc, but explain why these features are of interest.

We separated this sentence into two sentences, each of which contains the different features of the chloroplast genome sequencing, to explain the interest of each feature.

Lie 118: Indicate why you picked exactly those two species as outgroups. Both are very far away and thus it is not surprising that the phylogeny resolves these well.

We included the reason of our outgroup selection. Citing the thorough phylogenetic study on Rosaceae, we supported that Fragaria and Dasiphora are the closely related genera of the genus Rosa.

Line 152: Genome sized were nearly identical, differing by less than 400bps (Table 2)

This sentence was intended to be the transition point from the genome size to the genomic features. Because the genome sizes were specifically described in front part of this sentence, we decided not to describe the genome sizes again. For the natural transition, we simplified the sentence focusing on the general genomic features.

Line 172: There is no “confirmation” but rather “findings”. Given that only one individual was sequenced, it remains unknown how variable the chloroplast genomes are within a species.

We changed the word “confirmation” to “findings” followed your suggestion. In addition to this sentence, the other “confirmation” in the footnote of Table 2 is also replaced into “verification”. For the consistency, the materials and methods part was also slightly changed.

Line 263: They are marginally different at best

We added a new sentence following your suggestion, and changed the following sentences because the original sentences were awkward when it comes after the new sentence you have suggested.

Generally the discussion could include a statement about the broader impact of the results. Given the marginal difference among species as shown in Table 2 and in the phylogeny, the authors should discuss to which degree their species are actually species.

In the discussion part, now we emphasized that three Rosa sect. Synstylae species are geographically, ecologically, morphologically, and phenologically different. Based on these species characteristics, we suggested that they have a special evolutionary history rather than that they are actually to be one species. To discuss about species boundary, moreover, we suggested to analyze more nuclear data as well because the chloroplast data is not enough to discuss the species delimitation. We referred several similar case studies to Rosa sect. Synstylae to support Rosa sect. Synstylae species are distinct.

Reviewer 2 Report

The article is interesting  and useful. It presents three complete chloroplast genomes in the economically important genus Rosa, and it provides several markers that will be useful in future phylogenetic studies or as barcodes.

In my view, the paper is well written and clear. The introduction is complete and it really helps to understand and justify the aims of the paper.

The selection of materials is adequate, although the limited geographic range studied and the low number of samples may affect the authors’ conclusions (as they point out in the discussion section). The low number of specimens also renders the phylogenetic aims of the paper too ambitious.

The methods used are adequate and clearly described. The results section is well structured. The phylogenetic part of the discussion should be revised and some of its conclusions removed.

Comments:

Lines 20-22: 

This conclusion is not justified based on the results because only 3 species for section and four for subgenus is included

Section 2.3: 

The authors should indicate that sections Chinense and Indica are synonymous. Otherwise the connection between the text and the bibliography becomes confusing

Please justify why R. multiflora was used as reference genome. Outgroup selection should also be further explained in the text

Lines 158-159:

Please indicate the subgenus for Rosa odorata. 

Lines 166-169:

This sentence is confusing. Please re-write.

Lines 249-251:

Please ensure that these references [2, 5] support your assertion. Maybe new references should be included, if available.

Lines 252-254:

Please reconsider your phylogenetic results and make some corrections to figure 7. The relationships within the Synstylae are very poorly supported and nothing can be said about it. Besides, the number of specimens used is too low in order to establish the monophyly of the species. Finally, the Synstylae clade is not clear in the tree, and one of the nodes is barely visible. 

Lines 280-282:

Please, re-write. This sentence is confusing.

310-312:

There are other issues that might be obscuring the phylogenetic results. Rosa multiflora has been deemed polyphyletic and the definition of some of the taxa included in this article is not complete.

Author Response

Reviewer #2

The article is interesting and useful. It presents three complete chloroplast genomes in the economically important genus Rosa, and it provides several markers that will be useful in future phylogenetic studies or as barcodes.

In my view, the paper is well written and clear. The introduction is complete and it really helps to understand and justify the aims of the paper.

The selection of materials is adequate, although the limited geographic range studied and the low number of samples may affect the authors’ conclusions (as they point out in the discussion section). The low number of specimens also renders the phylogenetic aims of the paper too ambitious.

First, we delivered the phylogenetic results focusing on the accessions in this study. We avoided using the expression of taxonomic group levels (i.e., sectional levels or subgeneric levels) in the phylogenetic analyses. This might reduce the expression tendency of jumping into the general conclusion. On the other hand, we complemented the discussion on this problem. To support our discussion, we referred the previous Rosa phylogenetic studies and the other studies showing chloroplast sharing among closely related species. These studies will hopefully make our discussion reasonable.

The methods used are adequate and clearly described. The results section is well structured. The phylogenetic part of the discussion should be revised and some of its conclusions removed.

The phylogenetic part of the results and the discussion was intensively revised following your comments and suggestions.

Comments:

Lines 20-22: 

This conclusion is not justified based on the results because only 3 species for section and four for subgenus is included

We focused on the phylogenetic relationships of three Synstylae species, rather than the relationship of infrageneric groups.

Section 2.3: 

The authors should indicate that sections Chinense and Indica are synonymous. Otherwise the connection between the text and the bibliography becomes confusing

We indicated that Indicae is the synonym of Chinenses at where the name “Chinenses” was first used (line 33).

Please justify why R. multiflora was used as reference genome. Outgroup selection should also be further explained in the text

We justified the purpose of using R. multiflora as the reference in mVISTA analysis. We also included the reason of our outgroup selection. Citing the thorough phylogenetic study on Rosaceae, we supported that Fragaria and Dasiphora are the closely related genera of the genus Rosa.

Lines 158-159:

Please indicate the subgenus for Rosa odorata. 

We followed your suggestion. For consistency, the line 115 was revised as well.

Lines 166-169:

This sentence is confusing. Please re-write.

Not only this sentence, but we also changed the overall sentences related to this sentence for the clear delivery.

Lines 249-251:

Please ensure that these references [2, 5] support your assertion. Maybe new references should be included, if available.

This citation was inadequate as they supported only the species level relationships but not the sectional level relationships. We changed this sentence from the sectional level relationships to the species level relationships. Though species level relationships are corresponding to the result of reference [2,5], we decided not to referring these studies because we don’t have enough specimens to discuss on the correspondence to other studies.

Lines 252-254:

Please reconsider your phylogenetic results and make some corrections to figure 7. The relationships within the Synstylae are very poorly supported and nothing can be said about it. Besides, the number of specimens used is too low in order to establish the monophyly of the species. Finally, the Synstylae clade is not clear in the tree, and one of the nodes is barely visible. 

We focused on describing the phylogenetic relationships of species themselves, rather than of subgenera or sections, because we concluded that only one species couldn’t represent the taxomoic groups. Though the number of specimens is still low, we only suggested the monophyly of section Synstylae, not the species.  According to your review, we minimized describing the phylogenetic relationships in section Synstylae. Moreover, because the support values within section Synstylae were poor, the branch lengths were too short, and it was not sure where the arrow is pointing, we removed the support value annotation within section Synstylae.

Lines 280-282:

Please, re-write. This sentence is confusing.

We simplified this sentence to have a clear meaning.

310-312:

There are other issues that might be obscuring the phylogenetic results. Rosa multiflora has been deemed polyphyletic and the definition of some of the taxa included in this article is not complete.

We discussed on the polyphyly of R. multiflora and R. luciae, resulting in the possible existence of the distinct chloroplast haplotypes with these species. This will support the possibility of sampling bias problem. We also clarified the definition of species in this study with their distributions and ecological, morphological, and phenological characteristics in the introduction part. Following this, we also discussed on species boundaries because the phylogenetic analysis results concern the species identities.

Round 2

Reviewer 1 Report

I congratulate the authors for their nice revision. Particularly the map and the ecological details make this contribution of greater interest.

The only minor thing: Avoid short forms like "couldn't" in the Abstract.